# Indole-3-Propionic Acid, a Functional Metabolite of *Clostridium sporogenes*, Promotes Muscle Tissue Development and Reduces Muscle Cell Inflammation

**DOI:** 10.3390/ijms222212435

**Published:** 2021-11-18

**Authors:** Lei Du, Renli Qi, Jing Wang, Zuohua Liu, Zhenlong Wu

**Affiliations:** 1State Key Laboratory of Animal Nutrition, College of Animal Science and Technology, China Agricultural University, Beijing 100193, China; b20183040327@cau.edu.cn; 2Animal Nutrition Institute, Chongqing Academy of Animal Science, Chongqing 402460, China; qirl@cqaa.cn (R.Q.); wj57482199@163.com (J.W.); 3Key Laboratory of Pig Industry Sciences, Ministry of Agriculture, Chongqing 402460, China; 4Beijing Advanced Innovation Center for Food Nutrition and Human Health, China Agricultural University, Beijing 100193, China

**Keywords:** *C. sporogenes*, muscle inflammation, indole-3-propionic acid, miR-26a-2-3p, IL-1β

## Abstract

*Clostridium sporogenes* (*C. sporogenes*), as a potential probiotic, metabolizes tryptophan and produces an anti-inflammatory metabolite, indole-3-propionic acid (IPA). Herein, we studied the effects of *C. sporogenes* and its bioactive metabolite, IPA, on skeletal muscle development and chronic inflammation in mice. In the in vivo study, the muscle tissues and serum samples of mice with *C. sporogenes* supplementation were used to analyze the effects of *C. sporogenes* on muscle metabolism; the IPA content was determined by metabonomics and ELISA. In an in vitro study, C2C12 cells were exposed to lipopolysaccharide (LPS) alone or LPS + IPA to verify the effect of IPA on muscle cell inflammation by transcriptome, and the involved mechanism was revealed by different functional assays. We observed that *C. sporogenes* colonization significantly increased the body weight and muscle weight gain, as well as the myogenic regulatory factors’ (MRFs) expression. In addition, *C. sporogenes* significantly improved host IPA content and decreased pro-inflammatory cytokine levels in the muscle tissue of mice. Subsequently, we confirmed that IPA promoted C2C12 cells’ proliferation by activating MRF signaling. IPA also effectively protected against LPS-induced C2C12 cells inflammation by activating Pregnane X Receptor and restoring the inhibited miR-26a-2-3p expression. miR-26a-2-3p serves as a novel muscle inflammation regulatory factor that could directly bind to the 3′-UTR of IL-1β, a key initiator factor in inflammation. The results suggested that *C. sporogenes* with its functional metabolite IPA not only helps muscle growth development, but also protects against inflammation, partly by the IPA/ miR-26a-2-3p /IL-1β cascade.

## 1. Introduction

Sarcopenia is a common chronic muscle inflammation in seniors induced by lipotoxicity, in which host microbial imbalance causes the insufficient intake of protein and antioxidant nutrients [1]. The disturbances of protein synthesis and degradation in chronic muscle inflammation result in muscle atrophy and dysfunction [2,3]. Intestinal contents (for example, bacteria, foods, metabolic products) affect the inflammation of metabolic organs through the gut–organ axis [4,5,6]. Therefore, this study aimed to explore the effects of *C. sporogenes* and its metabolites on skeletal muscle development and inflammation.

The *Clostridium* genus is an important cluster of intestinal symbiotic bacteria, including *Clostridium* clusters IV and XIVa, which are necessary for the normalization of germ-free mice. The *Clostridium* genus is a potential probiotic, and its cellular components and metabolites, such as secondary bile acid, butyric acid, and indole-3-propionic acid (IPA), play a probiotic role by enhancing the intestinal barrier and interacting with the immune system [7]. As a member of the *Clostridium* genus, *C. sporogenes* synthesizes IPA via tryptophan catabolism and regulates the health of host intestinal cells and distal tissues [8,9]. IPA has been reported to regulate low-grade inflammation by enhancing mitochondrial oxidative stress activity and reducing cancer cell proliferation [10,11,12]. All evidence points to an anti-inflammatory effect of IPA; however, the molecular mechanisms involved are largely undetermined.

IPA is a tryptophan metabolite, and previous studies have indicated that amino acid administration is related to the regulation of microRNA (miRNA) expression in human skeletal muscle [13]. miRNAs are critical factors in regulating developmental and physiological cellular processes [14]. Accumulating evidence indicates that miRNAs act as control nodes for metabolic homeostasis [15,16]. For example, the dysregulation of miR-26a, miR-449a, and miR-30d is correlated with chronic inflammation [17,18,19]. However, whether specific miRNAs regulate the function of central metabolic organs in response to metabolite-derived cues is not clear. This might be a novel approach for elucidating the molecular mechanism by which IPA regulates inflammation and physiological functions by changing miRNA status.

In this study, we investigated the effects of *C. sporogenes* supplementation on metabolism and muscle inflammation in mice. The results demonstrated that *C. sporogenes* treatment altered aromatic amino acid (AAA) metabolism and improved inflammatory resistance in mice. The bioactivity of the tryptophan metabolite IPA was demonstrated, and we further investigated the possible mechanism by which IPA reduces muscle cell inflammation at the cellular and molecular levels. 

## 2. Results

### 2.1. C. sporogenes Supplementation Promoted Muscle Weight Gain by Increasing Myogenic Regulatory Factors in Mice

To explore the effects of *C. sporogenes* on skeletal muscle, *C. sporogenes* was gavaged at a dose of 1 × 10^8^ CFUs/200 μL for six weeks in mice, and the gavage was administered twice weekly (Figure 1A); the treatment methods refer to previous studies [7,20]. After that, we found that the colonization of the *Clostridium* cluster in the cecum of treated mice significantly increased in comparison with the control group (Figure 1B,C). *C. sporogenes* colonization effectively promoted body weight gain and increased the muscle weight in the front of the thigh (quadriceps), but had no significant effect on the feed intake (Figure 1D–G). Morphological sections showed that *C. sporogenes* colonization obviously increased the muscle fiber diameter and muscle cross-sectional area of the quadriceps (*p* ≤ 0.01; Figure 1H–J). Although the growth-promoting effect of *C. sporogenes* colonization on the muscle tissues of the lower leg (tibialis anterior muscle and gastrocnemius muscle) were not significant, we also measured the function development of gastrocnemius, which is important for muscle movement. Compared with the control group, the mRNA expression of myogenic regulatory factors, myocyte-specific enhancer factor 2D (MEF2D (1.13-fold), paired box 3 (Pax3) (1.70-fold), and Pax7 (1.82-fold) increased significantly. Meanwhile, myostatin (MSTN) (0.72-fold) was dramatically reduced (Figure 1K). It was demonstrated that *C. sporogenes* administration improved body weight and muscle weight gain and increased the mRNA expression of myogenic regulatory factors in muscle tissue.

### 2.2. C. sporogenes Altered AAA Metabolism and Reduced Pro-Inflammatory Factor Expression

In a subsequent study, we detected the effects of *C. sporogenes* on body metabolism. It has been reported that *C. sporogenes* is mainly related to AAA metabolism, especially tryptophan [21]. From the metabolomics analysis, we found that there were abundant AAA metabolites, including IPA, IAA, tryptamine, and tryptophan, in the supernatant of the *C. sporogenes* fermentation broth (Figure 2A). Concurrently, 702 metabolites were detected in mouse serum, and the PCA-3D diagram showed that the metabolism of mice changed significantly after *C. sporogenes* supplementation (Appendix A). There were 10 differential expression (DE) metabolites with VIP ≥ 1 and absolute log2 (fold change) ≥ 1 between groups (Appendix A). Consistent with the *C. sporogenes* supernatant, the KEGG enrichment analysis of the significant DE metabolites in mouse serum also points to AAA metabolism (Figure 2B), and their KEGG enriched pathways are shown in Appendix A. To identify the key AAA metabolites of the mouse serum, we expanded the DE metabolite screening criteria (VIP ≥ 1; absolute log2 (fold change) ≥ 1 is expanded to absolute log2 (fold change) ≥ 0.26). A total of 137 differential metabolites were screened, including 13 AAA metabolites (Appendix A). Ultimately, we found that tryptophan metabolites, such as shikimic acid, IAA, and indole sulfuric acid, were remarkably increased in mouse serum (fold change ≥ 1.2; Figure 2C). 

To determine whether the tryptophan metabolites changed simultaneously in muscle tissue, we measured the levels of representative tryptophan metabolites IPA, IAA, and kynurenine (KYN). Among them, IPA and IAA are mainly produced by bacterial tryptophan catabolism, while KYN is produced by the host’s own kynurenine pathway of tryptophan degradation [22]. *C. sporogenes* supplementation significantly increased the content of metabolites IPA and IAA and observably decreased KYN content in muscle (Figure 2D). It has been reported that KYN is a metabolite that is negatively correlated with muscle growth [23]. 

More importantly, we found that *C. sporogenes* colonization inhibited the mRNA expression of proinflammatory cytokines CCL2, IL-1β, TNFα, and NLRP3, and the expression of all the genes except NLRP3 was significantly different (0.74-, 0.57-, 0.65-, 0.79-fold, respectively; Figure 2E). Correlation analysis revealed that IPA was negatively correlated with inflammatory cytokines IL-1β and CCL2; IAA with CCL2 showed the same trend; while KYN with CCL2 showed the opposite trend (Figure 2F). Thus, we indicate that *C. sporogenes* significantly affected the AAA anabolism and produced anti-inflammatory substances IPA and IAA to inhibit proinflammatory cytokine expression, as well as reducing KYN content to promote muscle growth.

### 2.3. IPA, a Key Metabolite of C. sporogenes, Promoted Cell Proliferation and Alleviated C2C12 Cellular Inflammation Responses

Subsequently, we focused on the role of IPA in muscle cell proliferation and inflammation of C2C12 murine myoblasts. The results suggested that IPA at a low concentration of 0.1 mM remarkably increased myoblast cell viability (Figure 3A; *p* < 0.05) and promoted the expression of the myogenic regulatory factors, MEF2D (1.25-fold) and Myf5 (1.17-fold). IPA significantly inhibited MSTN (0.73-fold) expression in myoblasts (*p* < 0.05; Figure 3B). This suggested that 0.1 mM IPA promoted muscle cell proliferation by regulating the myogenic regulatory factor signals.

We evaluated the regulatory influence of IPA on LPS-induced muscle inflammation. C2C12 cells have two morphologies during development: myoblasts and myotubes (Appendix A). By comparison, we found that the differentiated C2C12 cells (myotubes) were more sensitive to LPS than myoblasts. The upregulation of pro-inflammatory factors’ expression such as CCL2, CCL5, and IL-1β was significantly higher than that in myoblasts after LPS treatment for 12 h (Appendix A; *p* < 0.05). Compared with the control group, myotubes treated with 1 μg/mL LPS for 12 h induced significant increases in the levels of four pro-inflammatory markers, CCL2, CCL5, IL-1β, and TNFα (Appendix A). Further, 1 μg/mL of LPS treatment led to a 38% reduction in myotube cell length (Appendix A). 

Different concentrations of IPA were used to alleviate the upregulation of pro-inflammatory markers induced by LPS treatment for 12 h (Appendix A), and 0.1 mM IPA pre-treated myotubes for 48 h significantly reduced the release of a key pro-inflammatory cytokine, IL-1β, in myotubes (Figure 3C,D). As the key receptor of IPA, the pregnane X receptor (PXR/NR1I2) regulates a series of exogenous or endogenous gene expressions involved in the inflammatory response after activation [24,25]. PXR has been shown to reduce proinflammatory cytokine secretion mainly by inhibiting the NF-κB signaling pathway [26,27]. Thus, the expression of PXR and the TLR4/MyD88/NF-κB signaling pathway proteins were detected in our study. It was found that IPA pretreatment effectively improved the inhibition of the receptor protein PXR and inflammatory proteins TLR4, MyD88, and NF-κB and inhibited the maturation of the ensuing gene IL-1β and the NLRP3 inflammasome (Figure 3E). Our results suggest that IPA also alleviates inflammation by PXR activation in muscle cells, causing the inhibition of the NF-κB signaling pathway and the secretion of proinflammatory cytokines.

### 2.4. Transcriptome Detection of Functional miRNAs Involved in Myotubular Inflammation

miRNAs are crucial for the progression and regulation of the inflammatory response [28]; therefore, we explored the major miRNAs involved in muscle cell inflammation by transcription to reveal the action mechanism of IPA. Overall, miRNA transcriptomes of myotubes obtained twenty-one DE miRNAs, with an FPKM value >10 and absolute log2 (fold change) ≥ 1, from the annotated genes (the threshold value was padj < 0.05), of which fifteen were downregulated and six were upregulated. The 15 downregulated DE miRNAs following LPS treatment are shown in the heatmap in Figure 4A. Furthermore, qPCR was used to verify the DE miRNAs. Results were consistent with the sequencing results, and the mRNA expression of miR-26a-2-3p (miR-26A), miR-30d-3p, miR-3061-5p, and miR-449a-5p in myotubes were significantly reduced with LPS treatment (*p* < 0.01; Figure 4B). Meanwhile, the targeted gene function enrichment of those DE miRNAs revealed 48 pathways (data shown in Appendix A; padj < 0.05), and the top 10 significant pathways are shown in a bubble chart. They include the Ras signaling pathway, PI3K-Akt signaling pathway, and Rap1 signaling pathway, which are involved in inflammation (Figure 4C). 

### 2.5. IPA Alleviated LPS-Induced Cellular Inflammatory Response by Regulating miRNAs’ Transcription

To determine the role of miRNAs in IPA anti-inflammation, the above anti-inflammatory miRNAs’ expression of gastrocnemius muscle tissue treated with *C. sporogenes,* and inflammatory myotubes treated with IPA were detected. Interestingly, it was confirmed that miR-26A (1.69-fold), miR-30d-3p (1.27-fold), and miR-449a-5p (1.18-fold) were significantly upregulated in the gastrocnemius of mice with *C. sporogenes* colonization (Figure 5A). IPA also significantly improved the downregulation of the four miRNAs induced by LPS in C2C12 cells (*p* < 0.05, Figure 5B). The more dramatically changed miR-26A was selected for further verification, which has been reported to be closely related to inflammation [29].

Next, we hypothesized that miR-26A overexpression could inhibit proinflammatory cytokine expression. Indeed, miR-26A mimics cotransfection, which significantly increased the expression of miR-26A mRNA in myotube cells and alleviated the inhibition of LPS on miR-26A (Figure 5C). That is, miR-26A overexpression significantly inhibited the inflammatory signaling pathway protein TLR4 and MyD88; moreover, miR-26A overexpression with LPS treatment significantly inhibited the TLR4 and NF-κB activation, as well as the excessive expression of downstream proinflammatory proteins TNFα and IL-1β (Figure 5D–E). Moreover, miR-26A overexpression remarkably suppressed the mRNA expression of pro-inflammatory makers, including the upregulation of NLRP3 (3.60 vs. 0.55, *p* < 0.01), IL-1β (2.24 vs. 0.49, *p* < 0.01), TNFα (1.64 vs. 0.83, *p* < 0.01), CCL2 (3.62 vs. 1.94, *p* < 0.01), and CCL5 (7.56 vs. 3.40, *p* < 0.001) induced by LPS in myotubes (Figure 5F). 

### 2.6. miR-26A Targeting the IL-1β mRNA 3′-UTR Alleviated Myotube Inflammation

The results of Targetscan prediction and inflammation-related pathway analysis revealed the candidate target genes of miR-26A, including IL-1β, PIK3R3, and SMAD2. qPCR results showed that mRNA levels of all screening target genes were significantly increased after LPS treatment, while the overexpression of miR-26A significantly inhibited them (Figure 6A). Finally, we intended to verify the targeting relationship between miR-26A and IL-1β, because of its high targeted score (Targetscan score = 94). Their correlation analysis also verified that miR-26A overexpression was significantly negatively correlated with IL-1β mRNA expression (*p* < 0.01; Figure 6B). 

The IL-1β mRNA 3′-UTR region was predicted to have an 8 mer pairing site seed match with miR-26A, and the binding sequence on IL-1β 3′-UTR was 276-A, 277-G, 278-A, 279-A, 280-C, 281-A, 282-G, and 283-A by Targetscan (Figure 6C). Then, we cloned the sequences containing target genes IL-1β 3′-UTR (WT) into the pGL3 Expression Reporter Vector System, which also encodes the reporter gene firefly luciferase. To verify whether IL1β is the target gene of miR-26A, we chose cotransfection into human renal epithelial cells (293T cells) with the miR-26A mimic. This was because of its high transfection efficiency and mature transfection conditions, which can exclude the influence of the operability of the experiment itself on the results. miR-26A’s mimicked cotransfection caused a marked increase in miR-26A expression and a decrease in luciferase transcription. Deletion of the predicted miR-26A final binding sequence on the IL-1β 3′-UTR formed a mutant sequence (MUT), which inhibited the miR-26A-induced decrease in transcriptional activity (Figure 6B–D). 

## 3. Discussion

In the present study, we are the first to report that *C. sporogenes* supplementation promotes muscle development and increased skeletal muscle weight in mice. The main action pathway is by changes in tryptophan metabolism and the production of the anti-inflammatory factor IPA. It is reported that *C. sporogenes* is involved in tryptophan metabolism [21], and the KEGG enrichment of the DE metabolite also points to AA metabolism in our study. Moreover, *C. sporogenes* supplementation significantly increased IPA while inhibiting KYN production in the mouse muscle tissues. KYN is a circulating tryptophan metabolite that is significantly associated with muscle atrophy and lipid peroxidation [30,31]. It is implicated in age-related diseases such as osteoporosis and inflammation [32]. Thus, KYN inhibition might contribute to mouse muscle weight gain and inflammatory resistance. 

This study also illustrated the possible mechanisms linking muscle inflammation involving IPA through miR-26A (Figure 7). IPA is a novel functional metabolite that functions as an anti-inflammatory and antioxidant and enhances barrier function or cytostatic bacterial activity [6,12]. IPA activates its receptor PXR by directly binding to the genomic regions of PXR or indirect crosstalk with other transcriptional factors (such as AHR or TLR4) that control many genes involved in transport, inflammation, cell apoptosis, and oxidative stress [33]; hence, we speculated that IPA works by activating PXR. It has been suggested that PXR might be a central regulator of IPA-mediated control of muscle cell inflammation. Studies have shown that PXR reduces proinflammatory cytokine secretion by inhibiting the NF-κB signaling pathway when inflammatory diseases occur [26,27]. Therefore, the IPA/PXR/NF-κB signaling pathway plays a role in the regulatory mechanism of muscle inflammation.

This study also investigated whether microbial metabolites could directly regulate miRNA gene expression and target inflammation. A previous study detected that decreased levels of miR-26a are correlated with increased chronic low-grade inflammation in obese mice [34]. Consistent with our observations, miR-26A inhibits several critical proinflammatory markers that control muscle inflammation caused by LPS. This suggests that the regulation of miR-26A might serve as a significant mechanism in muscle inflammation in response to microbiota-derived signals by IPA. miRNAs regulate target gene expression by degrading mRNA or inhibiting protein translation or degrading the polypeptides by binding complementarily to the 3′-UTR of their corresponding target genes [35]. Our data suggested that miR-26A alleviates inflammation by targeting IL-1β mRNA 3′-UTR transcription. This mechanism offers a potential way to trigger more durable changes in gene expression without the need for transcriptional or epigenetic regulation. 

Our study had several limitations. We found that tryptophan-derived metabolites reduced inflammation in myotubes, but future research needs to determine whether this phenomenon can be generalized in an inflammatory animal model and human cohorts. Concurrently, our research proposes that regulation of PXR and miR-26A overexpression by microbiota-derived signals is the key mechanism controlling host metabolic processes and muscle inflammation, but the causality was not well tested. 

## 4. Materials and Methods

### 4.1. Mouse Treatment

All experimental protocols involving animals were approved by the Chongqing Academy of Animal Science Animal Care and Use Committee. In the present study, 6-week-old male C57BL/6 mice (*n* = 8, Huafukang, Beijing, China) were maintained in groups of no more than 4 mice per cage with a stable controlled environment and free access to food and water. The mice were raised for one week without any intervention to adapt to the new environment and then divided into two groups randomly (Figure 1A): (1) a normal chow diet (Paddy, Chengdu, China) with bacterial culture media gavaging (NC group) and (2) a normal chow diet with *C. sporogenes* gavaging (CS group). Mice from the first group received sterile bacteria culture media as the vehicle, whereas the CS group mice were administered at twice a week intervals with *C. sporogenes* (American Type Culture Collection, ATCC^®^ 15579™) by oral gavage at a dose of 1 × 10^8^ CFUs/200 μL. The body weight of each mouse was measured once a week during the animal trial, which lasted for 6 weeks.

### 4.2. Bacterial Strains’ Culture and Cecal Clostridium Count

*C. sporogenes* was obtained from the American Type Culture Collection (ATCC^®^ 15579™) and cultured with thioglycolate medium and trypticase sulfite neomycin agar. The bacterial strain was usually incubated in an anaerobic atmosphere at 37 °C for 24–48 h.

The cecal contents of mice were dissolved in PBS at 1:1 (*w*/*v*), homogenized, and shaken, then placed in an 80 °C water bath for 10 min to kill nonspore-forming anaerobic bacteria. The sample mixtures were diluted to 10^3^, 10^4^, and 10^5^, respectively, and coated on tryptone sulfite cycloserine agar (TSC; Hopebio, HB0253, Qingdao, China), and then, the plates anaerobic incubated at 37 °C for 48 h to obtain the *Clostridium* anaerobic microbes that produce hydrogen sulfide. Cell counts of different intestinal segments and feces were determined by a microbial colony counter Interscience Scan^®^ 500 (Interscience, Saint Nom la Brétèche, France).

### 4.3. Hematoxylin and Eosin Staining 

Mice tibialis anterior were fixed at room temperature for 7 days in 4% paraformaldehyde. After dehydration in graded ethanol, diaphanization was performed with xylene, and then, the tissues were embedded in paraffin. Finally, the muscle tissue blocks were longitudinally cut into 3–4 μm sections along the muscle fibers on a rotary microtome Microm HM340E (Thermo Scientific, Waltham, MA, USA). For the histological study, the sections from 3 mice were stained by hematoxylin and eosin and examined by a Leica DM3000 microsystems (Leica, Milton Keynes, UK). Three regions were selected for each section, and the diameters of 5 muscle fibers were measured for each section. The analysis of the muscle fiber diameter and number was conducted using ImageJ software v1.8.0 (NIH, Bethesda, MA, USA).

### 4.4. Metabolome Sequencing and Data Analysis 

The serum samples and the supernatant of *C. sporogenes* fermentation broth were extracted by methanol (300 µL of pure methanol was added to 50 µL of samples), and then, the 150 μL of extracted supernatant was analyzed using an LC-ESI-MS/MS system (UPLC, ExionLC AD; MS, QTRAP^®^ System). The analytical conditions were as follows” UPLC: column, Waters ACQUITY UPLC HSS T3 C18 (1.8 μm, 2.1 mm * 100 mm); column temperature, 40 °C; flow rate, 0.4 mL/min; injection volume, 2 μL; solvent system, water (0.1% formic acid): acetonitrile (0.1% formic acid); gradient program, 95:5 *v*/*v* at 0 min, 10:90 *v*/*v* at 10.0 min, 10:90 *v*/*v* at 11.0 min, 95:5 *v*/*v* at 11.1 min, 95:5 *v*/*v* at 14.0 min.

Unsupervised principal component analysis (PCA) was performed by statistics function prcomp within R (www.r-project.org, accessed on 31 March 2021). The data were unit variance scaled before unsupervised PCA. Significantly regulated metabolites between groups were determined by the value of variable importance in projection (VIP) ≥ 1 and absolute log_2_ (fold change) ≥ 1. In order to avoid overfitting, a permutation test (200 permutations) was performed. Identified metabolites were annotated using the KEGG compound database (http://www.kegg.jp/kegg/compound/, accessed on 31 March 2021), and annotated metabolites were then mapped to the KEGG pathway database (http://www.kegg.jp/kegg/pathway.html, accessed on 31 March 2021). Pathways with significantly regulated metabolites mapped to them were then fed into metabolite sets enrichment analysis (MSEA), and their significance was determined by the hypergeometric test’s *p*-values. 

### 4.5. Enzyme-Linked Immunosorbent Assay

The tryptophan metabolite content of gastrocnemius tissue was measured with IPA, IAA, and KYN (Yiyan, china; EY-H554, EY-H669, EY-H8K1) according to the manufacturer’s protocols. The gastrocnemius tissue and normal saline were mixed in a ratio of 1:1 (*w*/*v*) and broken by ultrasound. Then, the supernatant was collected and measured with the tryptophan metabolite ELISA Kits and BCA protein assay kit (Cwbiotech, Beijing, China). The final concentration in the supernatants of the gastrocnemius tissue was normalized to the amount of the total protein of the tissues.

### 4.6. Cell Cultures and Treatment

All cells were obtained from the American Type Culture Collection (ATCC) and cultured with DMEM plus 10% heat-inactivated fetal bovine serum, penicillin (100 U/mL), and streptomycin (100 μg/mL), according to the ATCC’s recommendations. All cell culture reagents were from Thermo Scientific, and cells were grown at 37 °C in a 5% CO_2_ environment, while the medium was changed every 24 h.

C2C12 cells were grown to a confluency of approximately 80%, then the medium was changed to differentiation media composed of DMEM plus 2% horse serum to initiate differentiation. C2C12 cells were cultured 3 days in 6-well plates with differentiation medium, then pretreated with different concentrations of IPA (0.1, 0.25, and 0.5 mM) for 48 h, followed by 100 ng/mL and 1000 ng/mL LPS treatment for an additional 12 h and 24 h, respectively. Finally, cells were fixed or harvested for further analysis. 

C2C12 cells were grown to a confluency of approximately 80%, then the medium was changed to differentiation media composed of DMEM plus 2% horse serum to initiate differentiation. C2C12 cells were cultured 4 days in 6-well plates with differentiation medium, then transfected with an miR-26A mimic for the night, and the working concentration of the miR-26A mimic and mutant was each 20 nM. Finally, the cells were treated by 1000 ng/mL of LPS for an additional 12 h, then cells were fixed or harvested for further analysis.

The human renal epithelial cells 293T cells were grown for 12 h to a confluency of approximately 80%, then transfected with the miR-26A mimic or mutant for 24 h. The working concentration of expression plasmids was 0.4 μg/mL, and the working concentration of the miR-26A mimic and mutant was each 20 nM. 

### 4.7. Cell Viability Assay

C2C12 cells (10,000/well) were seeded in a 96-well plate to allow for cell adherence, then incubated with different concentrations of IPA (0.1, 0.25, and 0.5 mM) for 24 h, and the cell viability was determined by the Cell Counting Kit-8 (Beyotime, Shanghai, China).

### 4.8. Fluorescent Staining and Immunocytochemistry Analysis

Cells grown in 6-well plates were fixed with 4% paraformaldehyde solution. Fixed cells were permeabilized using a 0.2% Triton X-100 (Sangon Biotech, Shanghai, China) solution for 1 h. Following that, the cells were stained with Phalloidin (Solarbio, Beijing, China) to visualize the F-actin filaments in myotubes or incubated with rabbit anti-IL-1β polyclonal antibody (Proteintech, Wuhan, China) and Alexa Fluor secondary antibodies (Invitrogen, Thermo Scientific, Waltham, MA, USA) to evaluate the inflammatory levels. Hochest (Solarbio, Beijing, China) was used to counter-stain nuclei. All analyses were conducted using the ImageJ software.

For the myotube fusion index, the 3 sections of each group were randomly selected. Under the 100× microscope, 3–5 visual fields were randomly selected to calculate the ratio of the number of fused nuclei to the total number of nuclei.

### 4.9. cDNA Synthesis and RT-qPCR

RNA extraction and reverse transcription were performed as per the manufacturer’s instructions (Takara; No. 9109). Then, the resulting cDNA was subjected to real-time qPCR with gene-specific primers (primer sequences are listed in Appendix A). in the presence of TB Green^®^ Premix Ex Taq™ II using the StepOnePlus Real-Time PCR System (Applied Biosystems, Thermo Scientific, Waltham, MA, USA). Relative mRNA expression was determined using the 2^−ΔΔCT^ method with GAPDH or U6 serving as endogenous controls from the mouse samples.

### 4.10. RNA Sequencing and Data Analysis

The extracted high-quality RNA (RIN number > 7.0) was sequenced using the Illumina sequencing technology on an Illumina Hiseq 4000 (LC Bio, Hangzhou, China) according to the recommended protocols. The thresholds of p-adjusted (padj) < 0.05 were called differential expression (DE) genes. In order to better intuitively reflect the clustering expression pattern, log10 (norm value) was used to display the miRNA expression. At the same time, the norm value of differential miRNAs can also be displayed by the Z-score. Gene enrichment analysis was generated by DAVID (https://david.ncifcrf.gov/, accessed on 17 October 2020). KEGG pathway analysis was performed using the Cluster Profiler R package, and KEGG enrichment analysis is presented by the ggplot2, while the results were presented as a scatter plot. The screened differential miRNAs were predicted by the TargetScanMouse database (http://www.targetscan.org/vert_80/, accessed on 17 October 2020). The RNA-seq data were deposited into the Sequence Read Archive (SRA) with the Bioproject ID PRJNA 734955. 

### 4.11. Protein Extraction and Immunoblotting Assay

Total protein was extracted by RIPA (Beyotime, Shanghai, China) with PMSF (Solarbio, Beijing, China). Then, the supernatant was collected and measured with a BCA protein assay kit (Cwbiotech, Beijing, China). Protein was separated by SDS-polyacrylamide gel electrophoresis (Thermo Scientific, Waltham, MA, USA) and blotted onto polyvinylidene difluoride membranes (PVDF, Millipore, Billerica, MA, USA). Then, the PVDF membranes were blocked in 5% skim milk in TBS-T buffer at room temperature for 1 h, followed by incubation with primary antibody at 4 °C overnight. Then, the membranes were incubated with secondary antibodies at room temperature for 1 h. The antibodies used in the trial are listed in Appendix A. Protein bands were visualized by a Gel Image System (Bio-Rad, Hercules, CA, USA).

### 4.12. Correlation Analysis

Correlation analysis between muscle tryptophan metabolites and proinflammatory factors was calculated by the Pearson correlation algorithm, and we intuitively display the obtained numerical matrix through a heatmap. Data information in a two-dimensional matrix or table is reflected through color changes, and the color depth represents the size of the correlation coefficients (r-values).

### 4.13. Molecular Cloning Experiments and Dual Luciferase Assay

The miR-26a-2-3p mimic and mutant were custom synthesized by Nantong Biomics Biotechnology. The construction of the IL-1β 3′-UTR reporter was carried out using the pGL-3 basic luciferase reporter vector. The primers 5′ CTAGAGACATTAGGCAGCACTCTCTAGAACAGAACCTAGCTGTCAACGT 3′ and 5′ CTAGACGTTGACAGCTAGGT

TCTGTTCTAGAGAGTGCTGCCTAATGTCT 3′ were used for cloning. Then, luminescence was detected by the Dual-Glo^®^ Luciferase Assay System (Promega, Madison, WI, USA) according to the recommended protocols. The ratio of luminescence from the experimental reporter to luminescence from the control reporter was calculated.

### 4.14. Statistical Analysis

Data are shown as the means ± SEM. The significance of the difference was analyzed by the two-tailed Student’s *t*-test or ANOVA with post hoc tests, as indicated. All analyses were performed using GraphPad PRISM v.8.02 (GraphPad Software, San Diego, CA, USA). All results were considered statistically significant at a * *p* < 0.05, ** *p* < 0.01, and *** *p* < 0.001 vs. control cells and ^a^ # *p* < 0.05, ## *p* < 0.01, and ### *p* < 0.001 vs. LPS treated cells. 

## 5. Conclusions

In conclusion, our results demonstrated that *C. sporogenes* promotes muscle weight gain and regulates AAA metabolism to produce IPA, which contributes to protecting against inflammation. IPA alleviates myotube inflammation by activating PXR and elevating anti-inflammatory miRNAs, of which miR-26A targets the pro-inflammatory cytokine IL-1β 3′ UTR directly. This highlights that the IPA/miR-26A/IL-1β cascade is a promising therapeutic strategy for chronic muscle inflammation.

## Figures and Tables

**Figure 1 ijms-22-12435-f001:**
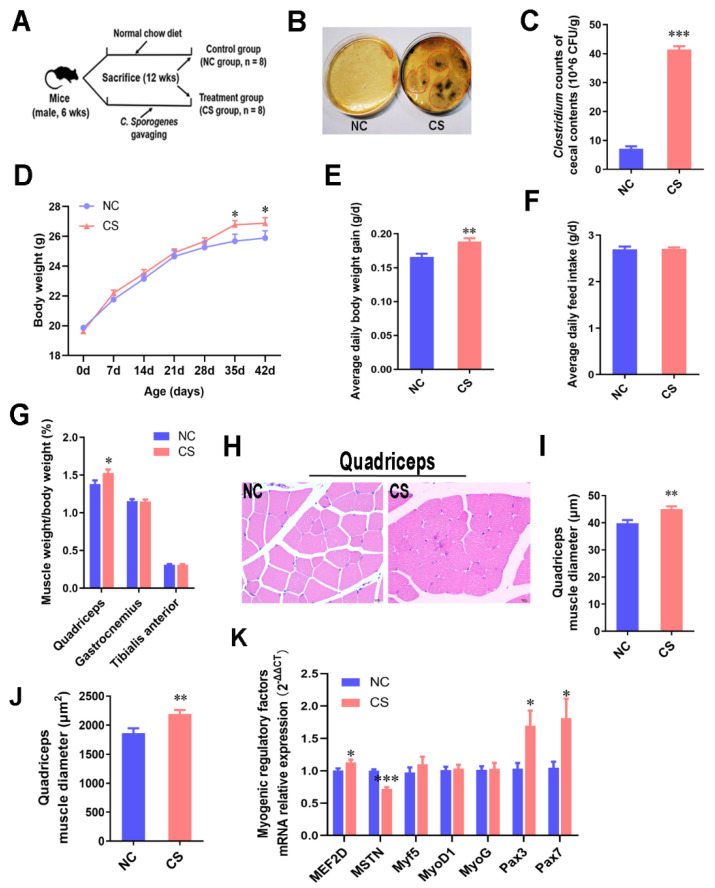
*C. sporogenes* supplementation promoted muscle weight gain by increasing myogenic regulatory factor signaling in mice. (**A**) Two groups of male C57BL/6 mice fed normal chow diet were administered bacteria culture media (NC group, *n* = 8) and *C. sporogenes* suspension, respectively (CS group, *n* = 8). (**B**,**C**) Plate counts of *Clostridium* genus colonization (black colonies circled in red) in the cecum after gavaging *C. sporogenes* for 42 days. (**D**) Body weights of C57BL/6 mice from 6 to 12 weeks of age. (**E**) Average body weight gain of the mice. (**F**) Average daily feed intake of the mice. (**G**) The ratio of muscle (gastrocnemius, quadriceps, and tibialis anterior) to the weight of mice. (**H**–**J**) HE staining of the quadriceps tissues in mice; the muscle fibers’ diameter (**I**) and muscle cross-sectional area (**J**) were measured. Scale bar = 10 μm. (**K**) The mRNA expression levels of myogenic regulatory factors (MEF2D, MSTN, Myf5, MyoD1, MyoG, Pax3, Pax7) in the gastrocnemius tissue treated as described above. The data shown are the means ± SEM, *n* = 8. * *p* < 0.05, ** *p* < 0.01, and *** *p* < 0.001. Unmarked graphs show no significant difference.

**Figure 2 ijms-22-12435-f002:**
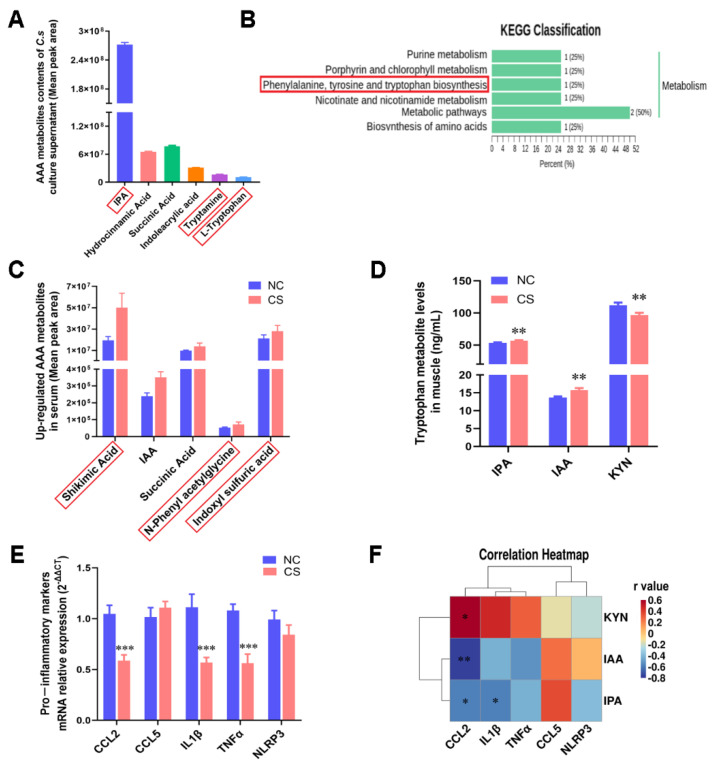
*C. sporogenes* altered AAA metabolism and reduced pro-inflammatory cytokine expression. (**A**) The top 6 AAA metabolites in *C. sporogenes* cell suspensions; the red box highlights the metabolites associated with tryptophan metabolism and the same below in this figure. (**B**) KEGG enrichment analysis of all differentially expressed metabolite. (**C**) The upregulation of key AAA metabolites in mice serum. (**D**) Levels of tryptophan metabolites IPA, IAA, and KYN in gastrocnemius tissues. (**E**) The mRNA levels of pro-inflammatory cytokines markers (CCL2, CCL5, IL-1β, TNFα, NLRP3) in the gastrocnemius tissue. (**F**) Correlation analysis of tryptophan metabolites (IPA, IAA, KYN) with pro-inflammatory cytokines (CCL2, CCL5, IL-1β, TNFα, NLRP3) in the gastrocnemius tissue. The data shown are the means ± SEM, *n* = 6. * *p* < 0.05, ** *p* < 0.01, and *** *p* < 0.001. Unmarked graphs show no significant difference.

**Figure 3 ijms-22-12435-f003:**
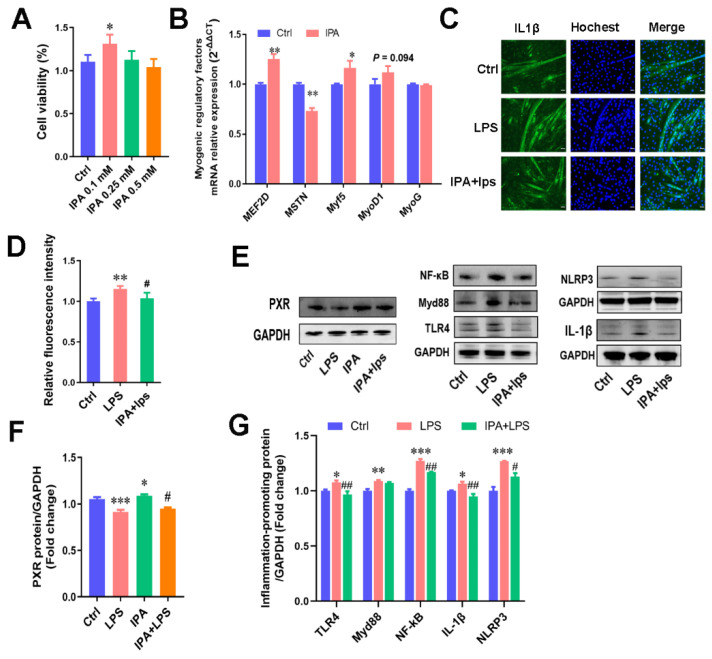
IPA promoted cells’ proliferation and alleviated inflammation responses in C2C12 cells. (**A**) Effects of different concentrations of IPA on the viability of myotube cells, which was detected by CCK8. Ctrl represents control cells, and IPA 0.1 mM represents cells treated with 0.1 mM IPA, and similarly hereinafter. (**B**) The mRNA expression levels of myogenic regulatory factors (MEF2D, MSTN, Myf5, MyoD1, MyoG) in m cells treated with 0.1 mM IPA. (**C**) Immunofluorescence staining of the pro-inflammatory cytokine IL-1β in myotube cells treated with LPS or LPS + 0.1 mM IPA. Scale bar = 50 μm. (**D**) The fluorescence gray value quantification. (**E**) The Western blot bands of proteins associated with inflammatory response (TLR4, MyD88, NF-κB, IL-1β, NLRP3) and the PXR receptor induced by IPA. (**F**) The gray value measurement of the PXR receptor. (**G**) The gray value measurement of the inflammatory response protein (TLR4, MyD88, NF-κB, IL-1β, NLRP3), inflammatory response protein by ImageJ. The data shown are the means ± SEM, *n* = 3. * *p* < 0.05, ** *p* < 0.01, *** *p* < 0.001 vs. control cells, and # *p* < 0.05, ## *p* < 0.01 vs. LPS treated cells. Unmarked graphs show no significant difference.

**Figure 4 ijms-22-12435-f004:**
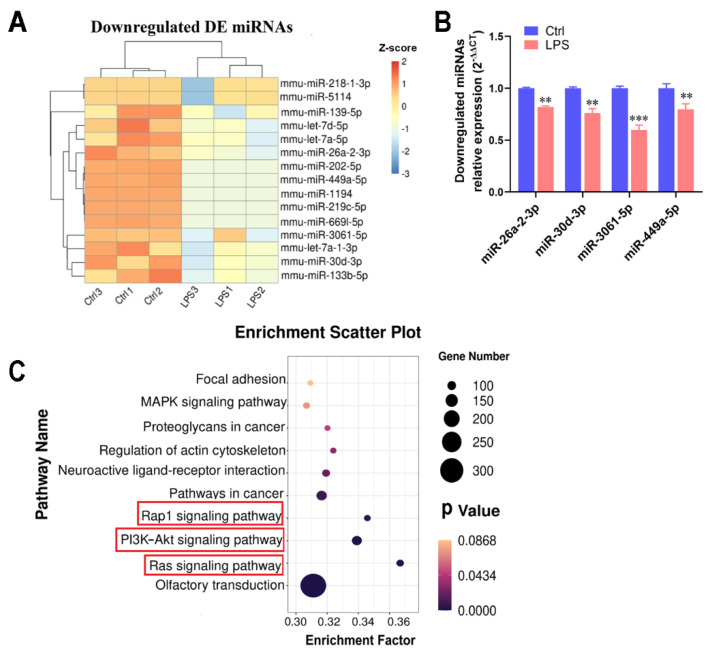
Transcriptome detection of functional miRNAs involved in myotubular cell inflammation. (**A**) Heatmap of 15 downregulated DE miRNAs in myotubes. (**B**) qPCR validation of the highly expressed DE miRNAs (FPKM value > 10 and absolute log2 (fold change) ≥ 1). (**C**) Bubble chart of the top 10 pathways through the targeted gene function enrichment; the red box highlights the pathway associated with inflammation. The data shown are the means ± SEM, *n* = 3. ** *p* < 0.01, and *** *p* < 0.001 vs. control cells. Unmarked graphs show no significant difference.

**Figure 5 ijms-22-12435-f005:**
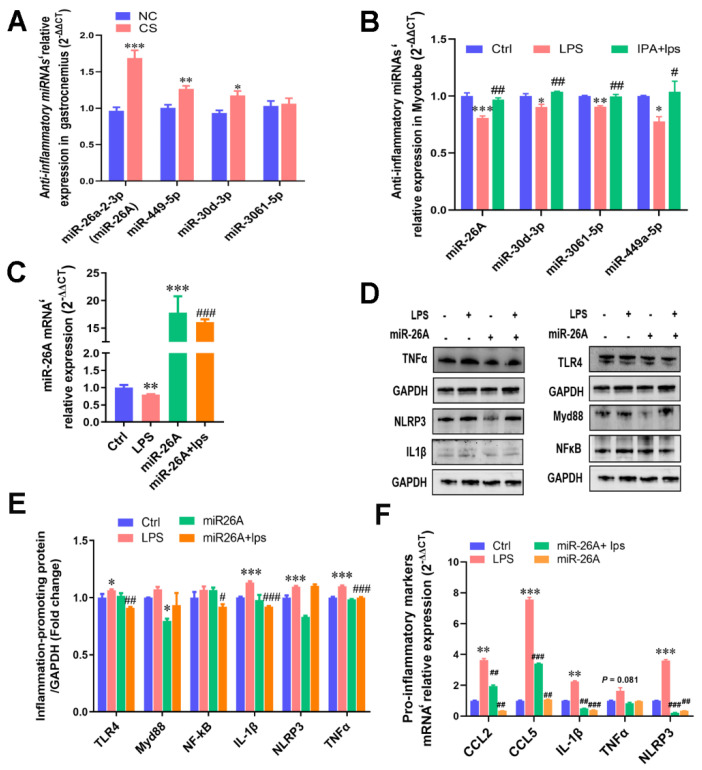
IPA alleviated LPS-induced cellular inflammatory response by interfering with anti-inflammatory miRNA transcription. (**A**) qPCR validation of the anti-inflammatory miRNAs in the gastrocnemius muscle tissue with *C. sporogenes* colonization. (**B**) Anti-inflammatory miRNAs’ expression levels of the inflammatory myotubes treated with different concentrations of IPA. (**C**) The miR-26a-2-3p (miR-26A) overexpression in myotubes. (**D**,**E**) The activation of miR-26A overexpression on inflammatory signaling pathways TLR4/MyD88/NF-κB and pro-inflammatory protein (NLRP3, TNFα, IL-1β) (**D**), as well as the protein gray value measured by ImageJ (**E**). (**F**) The suppression of miR-26A overexpression on pro-inflammatory markers’ mRNA expression in myotubes. The data shown are the means ± SEM, *n* = 3. * *p* < 0.05, ** *p* < 0.01, *** *p* < 0.001 vs. control cells and # *p* < 0.05, ## *p* < 0.01, and ### *p* < 0.001 vs. LPS treated cells. Unmarked graphs show no significant difference.

**Figure 6 ijms-22-12435-f006:**
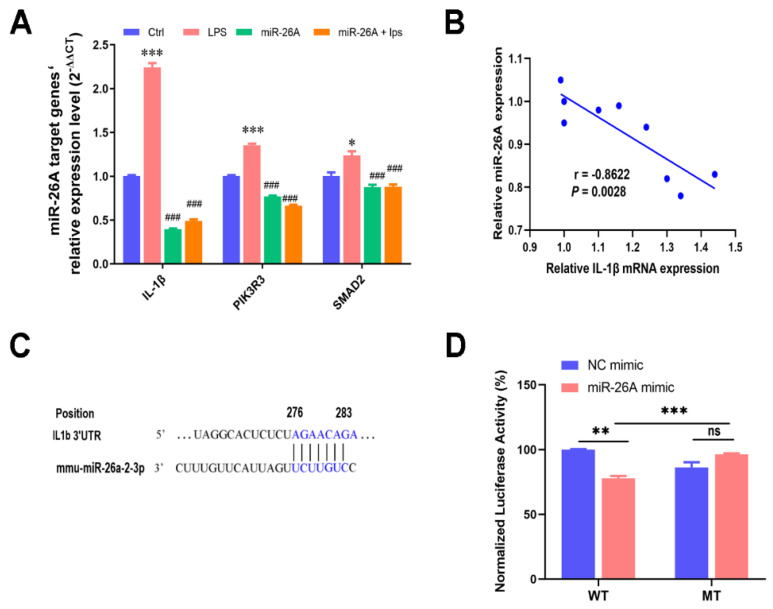
miR-26A targeting IL-1β mRNA 3′-UTR alleviated myotube inflammation. (**A**) qPCR validation of candidate target genes (IL-1β, Pik3R3, Smad2) of miR-26A. (**B**) Correlation analysis of miR-26A with IL-1β mRNA. (**C**) The binding region of miR-26A with the IL-1β mRNA 3′-UTR was predicted by Targetscan. (**D**) The dual luciferase assay verified the targeting relationship between miR-26A and IL-1β in the 293T cells. The data shown are the means ± SEM, *n* = 3. * *p* < 0.05, ** *p* < 0.01, *** *p* < 0.001 vs. control cells, and ### *p* < 0.001 vs. LPS treated cells. Unmarked graphs show no significant difference.

**Figure 7 ijms-22-12435-f007:**
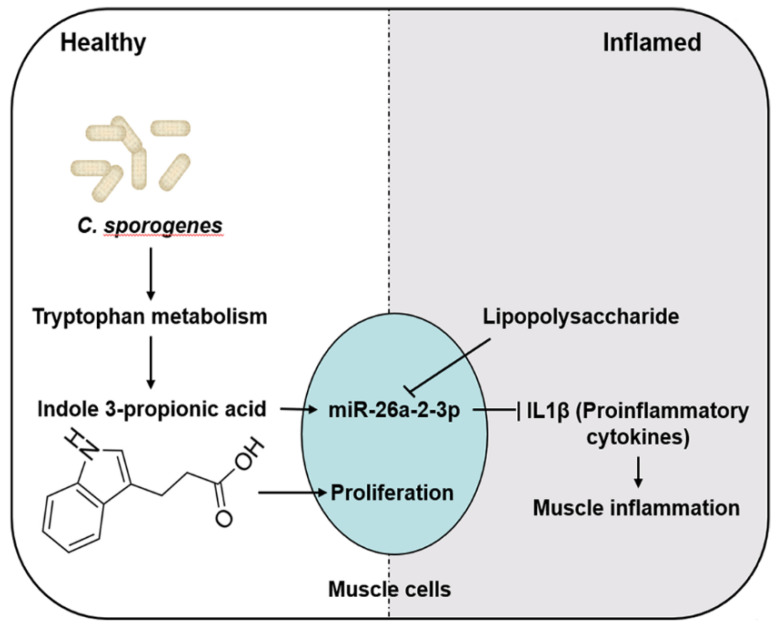
The potential molecular mechanisms of IPA-promoted muscle cell proliferation and reduced inflammation. Briefly, *C. sporogenes* produces anti-inflammatory metabolite IPA by changes in tryptophan metabolism. IPA promotes muscle cell proliferation by activating the myogenic regulatory factor signaling pathway. Moreover, IPA reduced cell inflammation and ensuing proinflammatory factors’ secretion through activating anti-inflammatory miR-26A targeting IL-1β in C2C12 cells. Those decreased proinflammatory cytokines from the muscle cells would reduce the occurrence of muscle inflammation.

## Data Availability

The authors declare the availability of all materials and data.

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
