# Peer review of "Indole-3-Propionic Acid, a Functional Metabolite of Clostridium sporogenes, Promotes Muscle Tissue Development and Reduces Muscle Cell Inflammation"

_ijms, 2021, doi:10.3390/ijms222212435_

Round 1

Reviewer 1 Report

Journal

IJMS (ISSN 1422-0067)

Manuscript ID

ijms-1406312

Type

Article

Title

Indole-3-propionic Acid, a Functional Metabolite of Clostridium sporogenes, Promotes Muscle Tissue Development and Reduces Muscle Cell Inflammation

Authors

Lei Du , Renli Qi , Jing Wang , Zuohua Liu * , Zhenlong Wu *

General considerations

In the manuscript entitled “Indole-3-propionic Acid, a Functional Metabolite of Clostridium sporogenes, Promotes Muscle Tissue Development and Reduces Muscle Cell Inflammation

” Du and colleagues show that Clostridium sporogenes diet supplementation provides some amelioration for skeletal muscle differentiation and inflammation. The authors sustain that this positive effect is due to Indole-3-propionic Acid, a tryptophan metabolite produced by C. sporogenes. Moreover, they show that IPA treatment is able to modulate miR-26A, involved in the control of pro-inflammatory cytokines in the skeletal muscle. Although the data are interesting, data presentation in the main text and figures is formally poor. The following suggestions and requests by this Reviewer aim at improving the reproducibility of results, the scientific soundness of the work and the overall presentation of data. Moreover, the figure are very low quality. Please increase the quality and adapt to A4 page.

Introduction

Lines 37-38: please fix “The disturbances of protein synthesis and degradation in chronic muscle inflammation, resulting in muscle atrophy and dysfunction” with “The disturbances of protein synthesis and degradation in chronic muscle inflammation results in in muscle atrophy and dysfunction”

Line 45: please spell “IPA” since it is the first appearance of this acronym.

Results and Material and methods

Figure 1A – lines 71-72: how did the authors determine the dose, frequency and time of treatment with C. -sporogenes? Please discuss this information in the manuscript

Figure 1B: the authors present a picture of agar plates with C. sporogenes colonies. There is no description of this procedure in the Material and Methods section. Please integrate with protocol about C. sporogenes culture conditions and protocol for the determination of CFU from cecum extract (including quantification of colonies).

Figure 1D: the authors show the daily increase of body weight. The authors should also show the body weight curve and the analysis between initial and final body weight. How was the daily increase calculated? Integrate this information in the Results and Materials and Methods section.

Figure 1E: the authors show the weight of quadriceps and gastrocnemius muscles while in figure F they show histology of tibialis anterior. The authors should also show the weight of this muscle, unless the choice of the particular muscle depot for histology is not justified.

Figure 1F: it is not clear whether the sections are transverse or longitudinal. The authors should provide this information in the Material and Methods section. Moreover, the authors should describe the pipeline analysis through the ImageJ software including how many sections per samples were analyzed. The authors should refer to peer-reviewed methods for cross-sectional analysis of skeletal muscle fibers. Alternatively, the authors should also perform immunofluorescence staining of cross-sections stained with an anti-laminin and a cross-sectional area analysis with frequency distribution of fiber caliber.

Figure 1I: the y axis lacks the indication of the units reported on the axis (fold change? 2^-DDCT?). please correct.

Lines 100-101: it is not clear which type of samples was processed for metabolomic analysis. The authors state about a “supernatant” but it is not clear in what this material consists of. Please provide this information in the appropriate section of results and materials and methods.

Figure S1B: please correct “insignificant” with “not significant”. Please specify what “variable importance projection” is. Please specify better in the materials and methods section the pipeline analysis for the metabolomic study.

Figure 2A,C,E: the y axis of these graphs lacks of the label of the unit. Please correct. For panel A and C, please specify in the figure legend what the red square represents.

Figure 2B: the authors should report a table with all the enrichment terms including adjusted p-value an justify the focus on amino acid metabolism-related terms.

Lines 107-108: the authors state they have “expanded the DE metabolite screening criteria”. This information is not clear. Which kind of expansion has been made? What are the differences between the expanded data and the previous ones? Please specify and discuss better how the analysis has been performed.

Line 120: the authors should not state that the supplementation with c sporogenes decreases nlrp3 gene expression since this decrease is not statistically significant. Please state this aspect in the result section.

Figure 2A: why the authors show only the top 6 metabolites? Please show all the DE metabolites with relevant fold change and p-value. Chose the appropriate form to report these data (Table, graph)

Line 140: change “skeletal muscle cells” with “murine myoblasts”.

Line 144: to state that IPA promote C2C12 cell proliferation the authors should perform a growth curve analysis and/or incorporation of BrdU showing the percentage of BrdU+ nuclei over total. This Reviewer suggests the authors to perform this experiment at low confluence and to perform at least 5 time point with calculation of the doubling time.

Line 148: to state that myotube are more sensitive that myoblasts the authors should perform a statistical test between myoblasts and myotube at the same concentration of LPS and show a p-value < 0.05 between the two differentiation states.

Figure S2 B-C: it is not clear what the control group is. in the graph a single control is shown for both myoblasts and myotubes. does the control group consist of myoblasts or myotubes? or maybe the control is separately myoblasts an myotubes for the two differentiation states? the authors should show the relative expression of both the controls. moreover, since this experiment is a time course, do the authors use a control for every time point?

Figure S2 D-E: instead of myotube lenght the authors should quantify the fusion index, that is a more standardized method more frequenty found in literature. please calculate te fusion index as the number of myonuclei per field normalized for the total number of nuclei per field.

Figure S2 F: at which time point this experiment has been perfomed. please ensure that the increase in gene expression of pro-inflammastory cytokines in sigure S2F is consistent with what is shown in figure S2B.

Figure 3 C-D: the increase in relative fluorescence intensity is not a consistent data since it can not be normalized. please perform a real time or western blot analysis to monitor Il1b expression under LPS or LPS+IPA. moreover, specify in the results section wheter the two molecules were used simultaneously or at different times of treatment (pre-treatment with IPA and then LPS, or vice versa?)

Figure 3E: the authors should show a graph reporting the densitometric quantitation of western blot including a statistical analysis and the number of independent experiments performed

Figure legend of Figure 3: “IPA promoted cells proliferation and alleviated inflammation responses in C2C12 cells”. since there is not an actual demonstration of the increase in cell proliferation, the authors should remove this statement from the caption of Figure 3.

Figure 4A: please specify whether the miRNA-Seq analysis has been performed on myotubes or myoblasts. Moreover, what is the scale unit of the color code of the heatmap? which kind of clustering analysis has been performed? please state this in the results and materials and methods section

Line 183-184: “that there was a large proportion of 183 the 48 pathways (padj < 0.05)” this part of the sentence is unclear. "a large proportion of the 48 pathways" at which pathways is referring to?

Figure 4B: the authors should report a table with all the significant pathways resulting from the analysis and the not limiting to the top 10. Moreover, change “rich factor” on the x axis with "enrichment factor". please specify the analysis protocol in the materials and methods section

Line 187: if these miRNAs are downregulated why the authors state about a fold change >2? please clarify also in the figure legend

Line 210: please specify that the experiment consists in miR-26A overexpression

Figure 5C: the authors should perform densitometric analysis of western blot and report graphs with statistical analysis including the number of independent replicates performed.

Lines 214-216 + Figure 5D: the data about Il6 are not reported in the graph while NLRP3 is reported in the graph but not in the text. please correct and justify in the rebuttal.

Line 229: please report in the materials and methods section the analysis with targetscan

Line 231: please write qPCR consistently across all the manuscript

Line 232: please modify "interference" with "treatment"

Line 234: please describe the correlation analysis in the materials and methods section

Line 242: the authors should show qPCR analysis to state that co-transfection of miR-26A mimic caused an increase in miR-26A expression

Figure 6D: the authors should show statistical p-value between NC mimic and miR-26A mimic in MT cells. moreover, the authors should justify the decrease in luciferase reported signal in NC mimic MT cells.

Discussion

Line 256: “… the proliferation of skeletal muscle cells…” Unless the authors directly show that C. sporogenes supplementation drives a higher cell proliferation, they should remove this statement from discussion.

Line 266: correct "illustrating" with "illustrates"

Line 269: The authors should rephrase this sentence since it seems that IPA itself binds to genomic regions

Line 270: Please mention these other transcriptional factors to give more information to the readership.

Line 277: Please change "this experiment" with "this study"

Lines 279-280: Please correct "chronic inflammation in the circulation of obese mice" with "chronic low-grade inflammation in obese mice".

Line 290: please fix “had” with “has”

Figure legend of Figure 7: please fix “anti-inflammation” with “anti-inflammatory”

Lines 313-314: 6 weeks old à 6-week-old

Line 325: please fix “anterior tibialis muscle” with “tibialis anterior”

Line 325: 4% paraformaldehyde à please specify time and temperature

Line 333: please specify how the sample extracts were generated

Lines 341-342: please report catalog numbers of elisa kits

Lines 342-344: how was the tissue manipulated? lysis buffer and protocol. why they normalized to DNA and not to protein content? how was DNA extracted and quantified? it has been done from the same lysate or the muscle has been minced in two or more part to perform the different assays?

Line 347: Does the cell culture medium of C2C12 cells contain glutamine? In the experience of this Reviewer C2C12 do not grow well without glutamine.

Line 371: The authors should specify the analysys protocol of fluorencence microscopy images

Lines 373-374: Please specify the catalog number of the product used.

Lines 390-391: Please fix this sentence, there is a repetition

Line 392: Please specify time and temperature of incubation for primary and secondary antibody and the blocking agent used.

Conclusions

Line 411-412: “In conclusion, our results demonstrate that C. sporogenes regulates tryptophan metabolism in mice, promotes muscle weight gain, and produces IPA, which contributes to inflammation.” Please fix this sentence since it is in discordance with the conclusion of the work

Author Response

Dear reviewer:

We thank the reviewer for the time and effort that they have put into reviewing the previous version of the manuscript. Their suggestions have enabled us to improve our work.

Based on the instructions provided, we uploaded the file of the revised manuscript. Accordingly, we have uploaded a copy of the original manuscript with all the changes highlighted by using the track changes mode in MS Word.

Appended to this letter is our point-by-point response to the comments raised by the reviewers. The comments are reproduced and our responses are given directly afterward in a different color (red).

We would like also to thank you for allowing us to resubmit a revised copy of the manuscript.

Sincerely,

Lei Du

Response to Reviewer 1 Comments

Point 1: The figure are very low quality. Please increase the quality and adapt to A4 page. 

Response 1: We have replaced these images throughout the text according to the comment. (Line 88, 135, 179, 205, 234, 265, 321; page 3 to page 11)

Point 2: Lines 37-38: please fix “The disturbances of protein synthesis and degradation in chronic muscle inflammation, resulting in muscle atrophy and dysfunction” with “The disturbances of protein synthesis and degradation in chronic muscle inflammation results in muscle atrophy and dysfunction”

Response 2: We have rephrased this sentence according to the comment (Line 39, page 1).

Point 3: Line 45: please spell “IPA” since it is the first appearance of this acronym.

Response 3: Thank you for your kind reminder, we have spelled IPA according to the comment (Line 47, page 2).

Point 4: Figure 1A – lines 71-72: how did the authors determine the dose, frequency and time of treatment with C. -sporogenes? Please discuss this information in the manuscript.

Response 4: The dose, frequency and time of treatment with C. sporogenes have referred to the original article in the manuscript, and through the preliminary experimental verification.  

Referring to the study of Madhukumar Venkatesh et al. (DOI: 10.1016/j.immuni.2014.06.014), C. sporogenes cultures at the logarithm growth period (12h) was selected to feed mice, at which the microbial count was 1×108 CFUs/mL. Moreover, the results of Susana Fuentes et al. (DOI: 10.1016/j.resmic.2008.02.005) have showed that interval feeding of microbiota was conducive to microbial colonization. Thus, the mice in our study have been gavaged for 6 weeks and twice weekly. A large number of Clostridium cluster were colonized in cecum of mice after graving microbial fluid, which also indicated the successful colonization of C. sporogenes (Line 75-76, page 2).

Point 5: Figure 1B: the authors present a picture of agar plates with C. sporogenes colonies. There is no description of this procedure in the Material and Methods section. Please integrate with protocol about C. sporogenes culture conditions and protocol for the determination of CFU from cecum extract (including quantification of colonies).

Response 5: Regarding the suggestion about the methods, we changed this part, the “Bacterial strains culture and count” section have been added in Material and Methods section 4.2 (Line 363-369, page 17).

Point 6: Figure 1D: the authors show the daily increase of body weight. The authors should also show the body weight curve and the analysis between initial and final body weight. How was the daily increase calculated? Integrate this information in the Results and Materials and Methods section.

Response 6:  We are grateful for the suggestion. We have added the Body weight curve, the Average daily feed intake histogram in Figure 1D and Figure 1F (Line 90, page 4), and the calculated methods have shown on the Materials and Methods section 4.1 (Line357-358, page 16).

Point 7: Figure 1E: the authors show the weight of quadriceps and gastrocnemius muscles while in figure F they show histology of tibialis anterior. The authors should also show the weight of this muscle, unless the choice of the particular muscle depot for histology is not justified.

Response 7:  Thank you for the suggestion, the weight of tibialis anterior has been added in Figure 1F (Line 88, page 3).

Point 8: Figure 1F: it is not clear whether the sections are transverse or longitudinal. The authors should provide this information in the Material and Methods section. Moreover, the authors should describe the pipeline analysis through the ImageJ software including how many sections per samples were analyzed. The authors should refer to peer-reviewed methods for cross-sectional analysis of skeletal muscle fibers. Alternatively, the authors should also perform immunofluorescence staining of cross-sections stained with an anti-laminin and a cross-sectional area analysis with frequency distribution of fiber caliber.

Response 8: We are grateful for the suggestion. We have added the sections analyse method in the Materials and Methods section 4.3 (Line 552-556, page 17). And the cross-sections and longitudinal-sections of skeleton muscle tissue are peer-reviewed methods (Xiaona Tian et al., DOI: 10.1021/acs.jafc.9b03040). In this study, the longitudinal section has been chosen to take into account the cross-sectional area measurement and the overall view of tibialis anterior muscle tissue. In addition, the observation of the whole segment of muscle fiber is convenient to accurately measure fiber diameter.

Point 9: Figure 1I: the y axis lacks the indication of the units reported on the axis (fold change? 2^-ΔΔCT?). please correct.

Response 9: We have modified the figures throughout the text according to the comment (Line 88, 135, 179, 205, 234, 265; page 3 to page 10).

Point 10: Lines 100-101: it is not clear which type of samples was processed for metabolomic analysis. The authors state about a “supernatant” but it is not clear in what this material consists of. Please provide this information in the appropriate section of results and materials and methods.

Response 10: We have modified this part in the Results section 2.2 (Line 108, page 4), and the Materials and Methods section 4.4 according to the comment (Line 382-382, page 17).

Point 11: Figure S1B: please correct “insignificant” with “not significant”. Please specify what “variable importance projection” is. Please specify better in the materials and methods section the pipeline analysis for the metabolomic study.

Response 11: We have changed the “insignificant” into “not significant” and we have changed the “variable importance projection” into “variable importance in projection” in the Figure S1B (Supplemental materials). It refers to VIP value. we have modified the analysis for the metabolomic study in the Materials and Methods section 4.4 (Line 391-395, page 17).

Point 12: Figure 2A,C,E: the y axis of these graphs lacks of the label of the unit. Please correct. For panel A and C, please specify in the figure legend what the red square represents.

Response 12: We have added the unit of the label in the figures throughout the text, and we have defined what the red box does in the figure legend (Line 141-142, page 7; Line 220-221, page 10).

Point 13: Figure 2B: the authors should report a table with all the enrichment terms including adjusted p-value an justify the focus on amino acid metabolism-related terms.

Response 13: We have added a table (Supplementary Table S1) to describe this result in the Supplementary Material (Line 114-115, page 4; Line 502, page 19).

Point 14: Lines 107-108: the authors state they have “expanded the DE metabolite screening criteria”. This information is not clear. Which kind of expansion has been made? What are the differences between the expanded data and the previous ones? Please specify and discuss better how the analysis has been performed.

Response 14: Thank you for your suggestion. I have described the screening conditions and results in detail in this part of the Results 2.2 (Line 116-118, page 4).

Point 15: Line 120: the authors should not state that the supplementation with c sporogenes decreases nlrp3 gene expression since this decrease is not statistically significant. Please state this aspect in the result section.

Response 15: Thank you for the suggestion, we have modified the sentence according to the comment (Line 130-131, page 5).

Point 16: Figure 2A: why the authors show only the top 6 metabolites? Please show all the DE metabolites with relevant fold change and p-value. Chose the appropriate form to report these data (Table, graph)

Response 16:  In Figure 2A, we have showed the content of aromatic amino acid metabolites in the C. sporogenes culture supernatant. Only the metabolites content of this sample was measured, so there was no comparison between two groups (Line139, page 5).

Point 17: Line 140: change “skeletal muscle cells” with “murine myoblasts”.

Response 17:  We rephrased this sentence according to the comment (Line 153, page 7).

Point 18: Line 144: to state that IPA promote C2C12 cell proliferation the authors should perform a growth curve analysis and/or incorporation of BrdU showing the percentage of BrdU+ nuclei over total. This Reviewer suggests the authors to perform this experiment at low confluence and to perform at least 5 time point with calculation of the doubling time.

Response 18: Thank you for underlining this deficiency. In our study, CCK-8 cell proliferation assay preliminarily proved that IPA promote the proliferation of myoblast. Although there is no in-depth study, such as the effect of different time points on the cell proliferation, has been carried out. Therefore, there would be more complete experiments to explore the mechanism of IPA promoting muscle cell proliferation in our next research. We will adopt the suggestions of reviewers in the following experiments.

Point 19: Line 148: to state that myotube are more sensitive that myoblasts the authors should perform a statistical test between myoblasts and myotube at the same concentration of LPS and show a p-value < 0.05 between the two differentiation states.

Response 19:  We have compared the difference in proinflammatory factors between myotube and myoblasts, as indicated in Figure S2B. This part of content also has been described in the Results section 2.3 (Line 162-164, page 7).

Point 20: Figure S2 B-C: it is not clear what the control group is. in the graph a single control is shown for both myoblasts and myotubes. does the control group consist of myoblasts or myotubes? or maybe the control is separately myoblasts an myotubes for the two differentiation states? the authors should show the relative expression of both the controls. moreover, since this experiment is a time course, do the authors use a control for every time point?

Response 20: This study focuses on the inflammatory level of cells, both myoblasts and myotubes are different growth states of the same cell. Theoretically, as a negative control, the inflammatory levels of myoblasts and myotubes (or myotubes at different points in time) should be the same (average relative expression levels = 1). Therefore, the inflammatory expression levels of the negative control cells were presented on one bar of the histogram in this study.

Point 21: Figure S2 D-E: instead of myotube lenght the authors should quantify the fusion index, that is a more standardized method more frequenty found in literature. please calculate te fusion index as the number of myonuclei per field normalized for the total number of nuclei per field.

Response 21: Thank you for underlining this deficiency. This section was revised and the cell fusion index graph in the Figure S2E, and the method have been added in the Materials and Methods section 4.8 (Line 436-438, page 18).

Point 22: Figure S2 F: at which time point this experiment has been perfomed. please ensure that the increase in gene expression of pro-inflammastory cytokines in sigure S2F is consistent with what is shown in figure S2B.

Response 22: Thank you for the suggestion, we have added the time point in the Results section 2.3, and we sure that the results have a consistent trend (Line 162-164, page 18).

Point 23: Figure 3 C-D: the increase in relative fluorescence intensity is not a consistent data since it can not be normalized. please perform a real time or western blot analysis to monitor Il1b expression under LPS or LPS+IPA. moreover, specify in the results section wheter the two molecules were used simultaneously or at different times of treatment (pre-treatment with IPA and then LPS, or vice versa?)

Response 23: Thank you for the suggestion, the method of relative fluorescence intensity normalized is referenced to the articles of Reich Maria et al. (DOI:10.1016/j.jhep.2021.03.029). We have added the western blot analysis graph in the Figure 3F-G (Line 184, page 8). We have described the treatment and time point in the Materials and Methods section 4.6 (Line 415-420, page 18).

Point 24: Figure 3E: the authors should show a graph reporting the densitometric quantitation of western blot including a statistical analysis and the number of independent experiments performed.

Response 24:  Thank you for the suggestion, we have added the graph of western blot densitometric quantitation in the Figure 3F-G (Line 184, page 8).

Point 25: Figure legend of Figure 3: “IPA promoted cells proliferation and alleviated inflammation responses in C2C12 cells”. since there is not an actual demonstration of the increase in cell proliferation, the authors should remove this statement from the caption of Figure 3.

Response 25: We have preliminarily confirmed that the proliferation effect of 0.25mM IPA on C2C12 cells by CCK-8 cell viability assay in Figure 3A (Line 184, page 8).

Point 26: Figure 4A: please specify whether the miRNA-Seq analysis has been performed on myotubes or myoblasts. Moreover, what is the scale unit of the color code of the heatmap? which kind of clustering analysis has been performed? please state this in the results and materials and methods section.

Response 26:  Thank you for the suggestion, we have added the unit of the color code in Figure 4A (Line 216, page 10), added the sample information in the Results section 2.4 (Line 199-200, page 9), and supplemented this method in the Materials and Methods section 4.10; Line 450-452, page 18).

Point 27: Line 183-184: “that there was a large proportion of 183 the 48 pathways (padj < 0.05)” this part of the sentence is unclear. "a large proportion of the 48 pathways" at which pathways is referring to?

Response 27:  Sorry, it's a written mistake, we have modified the sentence (Line 207-208, page 9).

Point 28: Figure 4B: the authors should report a table with all the significant pathways resulting from the analysis and the not limiting to the top 10. Moreover, change “rich factor” on the x axis with "enrichment factor". please specify the analysis protocol in the materials and methods section.

Response 28: Thank you for the suggestion, we have added the data in the Supplementary Table S3 (Line 502, page 19). And we have changed the phrase of “rich factor” in the Figure 4C (Line 215, page 10), added the analysis protocol in the Materials and Methods section 4.10 (Line 454-456, page 18).

Point 29: Line 187: if these miRNAs are downregulated why the authors state about a fold change >2? please clarify also in the figure legend.

Response 29:  Thank you for your suggestion. This description is not rigorous enough and we have revised it into “absolute log2 (fold change) ≥ 1” throughout the text (Line 111,116-117, page 4; Line 200, page 9; Line 219, page 10; Line 394, page 17).

Point 30: Line 210: please specify that the experiment consists in miR-26A overexpression.

Response 30: We have noted it in the Results section 2.5 and the figure legend (Line 236-238, page 11; Line 253, page 12).

Point 31: Figure 5C: the authors should perform densitometric analysis of western blot and report graphs with statistical analysis including the number of independent replicates performed.

Response 31: We have added the western blot analysis graph in the Figure 5E, and the number of independent replicates is shown on the figure legend (Line 257-258, page 12-13).

Point 32: Lines 214-216 + Figure 5D: the data about Il6 are not reported in the graph while NLRP3 is reported in the graph but not in the text. please correct and justify in the rebuttal.

Response 32: Sorry, it's a written mistake, we have changed it (Line 244, page 11).

Point 33: Line 229: please report in the materials and methods section the analysis with targetscan.

Response 33:  Thank you for the suggestion, we have added it in the Materials and Methods section 4.10 (Line 455-456, page 18).

Point 34: Line 231: please write qPCR consistently across all the manuscript.

Response 34: We have modified this expression throughout the text according to the comment (Line 263, page 13; Line 441, page 18).

Point 35: Line 232: please modify "interference" with "treatment".

Response 35: We have changed it according to the comment (Line 264, page 13).

Point 36: Line 234: please describe the correlation analysis in the materials and methods section.

Response 36: Thank you for the suggestion, we have added this method in the Materials and Methods section 4.12 (Line 470-475, page 19).

Point 37: Line 242: the authors should show qPCR analysis to state that co-transfection of miR-26A mimic caused an increase in miR-26A expression.

Response 37:  Thank you for the suggestion, we have added this results in the Figure 5C and Results 2.5 (Line 248, page 12; Line 236-237, page 11).

Point 38: Figure 6D: the authors should show statistical p-value between NC mimic and miR-26A mimic in MT cells. moreover, the authors should justify the decrease in luciferase reported signal in NC mimic MT cells.

Response 38: We have added significance analysis in Figure 6D. Statistical results of NC mimic and miR-26A mimic in MT cells showed that P value = 0.231, and there was not significant (Line 282, page14).

Point 39: Line 256: “… the proliferation of skeletal muscle cells…” Unless the authors directly show that C. sporogenes supplementation drives a higher cell proliferation, they should remove this statement from discussion.

Response 39: Thank you for underlining this deficiency. This section was revised and modified according to the results presented in this work (Line 289-290, page 14).

Point 40:  Line 266: correct "illustrating" with "illustrates".

Response 40:  We have changed it according to the suggestion (Line 300, page 15).

Point 41: Line 269: The authors should rephrase this sentence since it seems that IPA itself binds to genomic regions.

Response 41:  We rephrased this sentence according to the comment (Line 303-304, page 15).

Point 42: Line 270: Please mention these other transcriptional factors to give more information to the readership.

Response 42: Thank you for the suggestion, we have added it according to the suggestion (Line 305, page 15).

Point 43: Line 277: Please change "this experiment" with "this study" .

Response 43: We rephrased this sentence according to the comment (Line 312, page 15).

Point 44: Lines 279-280: Please correct "chronic inflammation in the circulation of obese mice" with "chronic low-grade inflammation in obese mice".

Response 44: We rephrased this sentence according to the comment (Line 314-315, page 15).

Point 45: Line 290: please fix “had” with “has”.

Response 45:  We have changed it according to the suggestion (Line 325, page 15).

Point 46: Figure legend of Figure 7: please fix “anti-inflammation” with “anti-inflammatory”.

Response 46: We have changed it according to the suggestion (Line 340, page 16).

Point 47: Lines 313-314: 6 weeks old à 6-week-old.

Response 47:  We rephrased this sentence according to the comment (Line 349-350, page 16).

Point 48: Line 325: please fix “anterior tibialis muscle” with “tibialis anterior”.

Response 48: We have changed it according to the suggestion (Line 372, page 17).

Point 49: Line 325: 4% paraformaldehyde à please specify time and temperature.

Response 49: We have added it according to the suggestion (Line 372-373, page 17).

Point 50: Line 333: please specify how the sample extracts were generated.

Response 50: Thank you for the suggestion. We have added the information required as explained above (Lines 382-384, page 17).

Point 51: Lines 341-342: please report catalog numbers of elisa kits.

Response 51:  We have added the information required as explained above (Lines 403-406, page 17).

Point 52: Lines 342-344: how was the tissue manipulated? lysis buffer and protocol. why they normalized to DNA and not to protein content? how was DNA extracted and quantified? it has been done from the same lysate or the muscle has been minced in two or more part to perform the different assays?

Response 52:  This is a clerical error. It does measure the protein concentration (Lines 408, page 17), and it has been done from the same lysate. We have supplemented the lysis liquid preparation and concentration determination protocol in the Materials and Methods section 4.5 (Lines 404-406, page 17).

Point 53: Line 347: Does the cell culture medium of C2C12 cells contain glutamine? In the experience of this Reviewer C2C12 do not grow well without glutamine.

Response 53:  Yes, as the reviewer said. We use the high- glucose DMEM culture medium with 862 mg/L L-glutamine.

Point 54: Line 371: The authors should specify the analysys protocol of fluorencence microscopy images.

Response 54:  Thank you for the suggestion. We have added the information in the Materials and Methods section 4.8 (Line 436-438, page 18).

Point 55: Lines 373-374: Please specify the catalog number of the product used.

Response 55: We have added the information required as above (Line 441, page 18).

Point 56: Lines 390-391: Please fix this sentence, there is a repetition.

Response 56:  We rephrased this sentence according to the comment (Line 462, page 19).

Point 57: Line 392: Please specify time and temperature of incubation for primary and secondary antibody and the blocking agent used.

Response 57: We have added the information required as explained above (Line 464-467, page 19).

Point 58: Line 411-412: “In conclusion, our results demonstrate that C. sporogenes regulates tryptophan metabolism in mice, promotes muscle weight gain, and produces IPA, which contributes to inflammation.” Please fix this sentence since it is in discordance with the conclusion of the work.

Response 58:  We have modified the sentence according to the comment (Line 492-494, page 19).

Reviewer 2 Report

Indole-3-propionic Acid, a Functional Metabolite of Clostridium sporogenes, Promotes Muscle Tissue Development and Reduces Muscle Cell Inflammation

Du et. al., IJMS

The authors present a study of the effects of Indole-3-propionic Acid (IPA) on the development of skeletal muscle and the inhibition of the inflammatory response therein.  This is presented in the context of IPA being released from cultures of Clostridium sporogenes within the gut.  The authors conduct metabolomic, transcriptomic, and protein analyses to show the beneficial effects of IPA for the mouse. This is a well-designed and conducted study, from which appropriate and straightforward conclusions are drawn.  However, several aspects in the introductory and concluding experiments create confusion for the logical flow.  Moderate attention also needs to be paid to English grammar. Specific recommendations for improving the manuscript prior to publication are listed below.

Line 21: This is the first instance of LPS.  Please define here before continuing to use the acronym

Line 27: The verb is missing from this sentence.  Should it read “IPA also effectively protected against…”?

Line 31: Again, the verb is missing.

Line 72: For clarity, please add here that the gavage was administered twice weekly, as is stated in the Materials and Methods section

Line 74: Please provide information in the Materials and Methods section regarding procedures used to culture this bacteria or confirm that they are members of the Clostridium genus.

Line 76: Was weekly or biweekly body weight measured? It would be helpful to see the full time course curve of body weight gain instead of just average daily weight gain.

Line 241: It is entirely unclear why HEK293 cells were used in these experiments.  The entirety of the manuscript has focused upon skeletal muscle in mice, and thus the use of HEK293 cells for testing the effect of miR-26a on inflammatory markers seems illogical here. Why would C2C12 lines not be used here as well?  The physiological implications of IPA on the Kidney or in humans are not discussed in the context of the rest of the experiments, and thus this entire figure (figure 6) and group of experiments is out of place.  It is also unclear from figure 6B which samples were treated with the overexpression vector and which were not.

Author Response

Dear reviewer:

We thank the reviewer for the time and effort that they have put into reviewing the previous version of the manuscript. Their suggestions have enabled us to improve our work.

Based on the instructions provided, we uploaded the file of the revised manuscript. Accordingly, we have uploaded a copy of the original manuscript with all the changes highlighted by using the track changes mode in MS Word.

Appended to this letter is our point-by-point response to the comments raised by the reviewers. The comments are reproduced and our responses are given directly afterward in a different color (red).

We would like also to thank you for allowing us to resubmit a revised copy of the manuscript.

Sincerely,

Lei Du

Response to Reviewer 2 Comments

Point 1: Line 21: This is the first instance of LPS.  Please define here before continuing to use the acronym.

Response 1: We have completed the full name of LPS according to the comment (Line 21, page 1).

Point 2: The verb is missing from this sentence.  Should it read “IPA also effectively protected against…”? Line 31: Again, the verb is missing.

Response 2: Thank you for the suggestion, we have added the verb (Line 27, 31; page 1).

Point 3: Line 72: For clarity, please add here that the gavage was administered twice weekly, as is stated in the Materials and Methods section

Response 3: We are grateful for the suggestion. The sentence “and the gavage was administered twice weekly” were added according to the comment (Line 75, page 2).

Point 4: Line 74: Please provide information in the Materials and Methods section regarding procedures used to culture this bacteria or confirm that they are members of the Clostridium genus.

Response 4: We have added this protocol in the Material and Methods section 4.2 according to the comment (Line 360-370, page 17).

Point 5: Line 76: Was weekly or biweekly body weight measured? It would be helpful to see the full time course curve of body weight gain instead of just average daily weight gain.

Response 5: Thank you for the suggestion, we have added the body weight curve in Figure 1D (Line 89, page 3).

Point 6: Line 241: It is entirely unclear why HEK293 cells were used in these experiments.  The entirety of the manuscript has focused upon skeletal muscle in mice, and thus the use of HEK293 cells for testing the effect of miR-26a on inflammatory markers seems illogical here. Why would C2C12 lines not be used here as well?  The physiological implications of IPA on the Kidney or in humans are not discussed in the context of the rest of the experiments, and thus this entire figure (figure 6) and group of experiments is out of place. It is also unclear from figure 6B which samples were treated with the overexpression vector and which were not.

Response 6: In the Dual luciferase reporter assay, we aimed to verify whether IL1β is the target gene of miR-26A. The reasons of we chose 293T cells are their high transfection efficiency and mature transfection conditions, which can exclude the influence of the operability of the experiment itself on the results.

Round 2

Reviewer 1 Report

Journal

IJMS (ISSN 1422-0067)

Manuscript ID

ijms-1406312

Type

Article

Title

Indole-3-propionic Acid, a Functional Metabolite of Clostridium sporogenes, Promotes Muscle Tissue Development and Reduces Muscle Cell Inflammation

Authors

Lei Du , Renli Qi , Jing Wang , Zuohua Liu * , Zhenlong Wu *

General considerations

The Authors have accomplished all the points raised by this Reviewer. Some minor points are requested below:

  1. In figure 1G the authors show that quadriceps, but not gastrocnemius or tibialis anterior muscles, gains weight upon supplementation with CS. The histology showing section and CSA analysis of tibialis anterior is not justified. Moreover, can the authors discuss why they observe an increase in CSA of tibialis anterior muscles without changes in fiber number and muscle weight? If there is not hyperplasia (increase of fiber number) there should be hypertrophy (increase in fiber size). So the weight of the muscle would be higher. The authors can also show in supplementary material the raw weight of the three muscles without normalization.
  2. In the revised versione the authors report Supplementary Table S1 to show the enrichment values of KEGG pathways. In line 114 the authors state that such pathways are significantly enriched but in table S1 none KEGG report a p-value < 0.05. The authors should state that the enrichment was not significant but they focused on these pathways for correlation with previous data.
  3. Lines 162-164: the point was sufficiently corrected but the sentence refers both times to myoblasts. Maybe it is a refuse

Author Response

Point 1: In figure 1G the authors show that quadriceps, but not gastrocnemius or tibialis anterior muscles, gains weight upon supplementation with CS. The histology showing section and CSA analysis of tibialis anterior is not justified. Moreover, can the authors discuss why they observe an increase in CSA of tibialis anterior muscles without changes in fiber number and muscle weight? If there is not hyperplasia (increase of fiber number) there should be hypertrophy (increase in fiber size). So the weight of the muscle would be higher. The authors can also show in supplementary material the raw weight of the three muscles without normalization.

Response 1: We are grateful for the suggestion. In accordance with the reviewer concerns, we decide to do the paraffin section of quadriceps tissues. Results showed that the CS addition have a better effect of promoting muscle growth in quadriceps. So the results of this part have been replaced in the paper (Line 77-91, page 2-3).

Point 2: In the revised versione the authors report Supplementary Table S1 to show the enrichment values of KEGG pathways. In line 114 the authors state that such pathways are significantly enriched but in table S1 none KEGG report a p-value < 0.05. The authors should state that the enrichment was not significant but they focused on these pathways for correlation with previous data.

Response 2: Thank you for the suggestion, it's a written mistake. What is significant here should be the differential metabolites, while KEGG enrichment is only the metabolic pathways involved in these differential metabolites. We have modified the sentence according to the comment (Line 113-115, page 4).

Point 3: Lines 162-164: the point was sufficiently corrected but the sentence refers both times to myoblasts. Maybe it is a refuse.

Response 3:  Thank you for the suggestion, we have modified the sentence according to the comment (Line 163, page 6).

Reviewer 2 Report

The authors have adequately addressed the vast majority of the reviewers’ concerns.  Regarding the use of HEK293 cells in figure 6: since the intention of this experiment is solely to determine an interaction between miR-26a and IL-1β, the rationale behind the use of these cells in place of C2C12 cells is logical.  However, because of the abrupt switch in focus from skeletal muscle throughout the rest of the manuscript, the authors should explicitly state in this paragraph the rationale for using these cells instead of C2C12 cells and indicate that it is solely for the purpose of identifying the interaction. 

Author Response

Point 1: The authors have adequately addressed the vast majority of the reviewers’ concerns.  Regarding the use of HEK293 cells in figure 6: since the intention of this experiment is solely to determine an interaction between miR-26a and IL-1β, the rationale behind the use of these cells in place of C2C12 cells is logical.  However, because of the abrupt switch in focus from skeletal muscle throughout the rest of the manuscript, the authors should explicitly state in this paragraph the rationale for using these cells instead of C2C12 cells and indicate that it is solely for the purpose of identifying the interaction.

Response 1: We are grateful for the suggestion. The reasons have added in the Results section 2.6 (Line 262-266, page 9) and the figure legend of Figure 6D (Line 277, page 9).